# NEURO-COVAX: An Italian Population-Based Study of Neurological Complications after COVID-19 Vaccinations

**DOI:** 10.3390/vaccines11101621

**Published:** 2023-10-21

**Authors:** Maria Salsone, Carlo Signorelli, Alessandro Oldani, Valerio Fabio Alberti, Vincenza Castronovo, Salvatore Mazzitelli, Massimo Minerva, Luigi Ferini-Strambi

**Affiliations:** 1Institute of Molecular Bioimaging and Physiology, National Research Council, 20125 Milan, Italy; 2Sleep Disorders Center, Division of Neuroscience, San Raffaele Scientific Institute, 20127 Milan, Italy; 3School of Medicine, Vita-Salute San Raffaele University, 20132 Milan, Italy; 4Sovrintendenza Sanitaria del Gruppo San Donato, 20122 Milan, Italy; alberti.fabio@grupposandonato.it; 5Direzione Sanitaria, San Raffaele Scientific Institute, 20127 Milan, Italy; 6Sleep Disorders Center, Division of Neuroscience, Vita-Salute San Raffaele University, 20132 Milan, Italy

**Keywords:** BNT162b2 vaccine, mRNA-1273 vaccine, ChAdOx1nCoV-19 vaccine, Ad26.COV2.S vaccine, neurological adverse events, COVID-19 infection

## Abstract

Objective: In this Italian population-based study, we aimed to evaluate the neurological complications after the first and/or second dose of COVID-19 vaccines and factors potentially associated with these adverse effects. Methods: Our study included adults aged 18 years and older who received two vaccine doses in the vaccination hub of Novegro (Milan, Lombardy) between 7 and 16 July 2021. The NEURO-COVAX questionnaire was able to capture the neurological events, onset and duration. That data that were digitized centrally by the Lombardy region were used to match the demographic/clinical characteristics and identify a vulnerability profile. Associations between vaccine lines and the development of complications were assessed. Digital healthcare system matching was also performed to evaluate severe neurological complications (Guillain-Barrè syndrome, Bell’s palsy, transverse myelitis, encephalitis) and the incidence of hospital admissions and/or the mortality rate after two doses of the vaccines. Results: The NEURO-COVAX-cohort included 19.108 vaccinated people: 15.368 with BNT162b2, 2077 with mRNA-1273, 1651 with ChAdOx1nCov-19, and 12 with Ad26.COV2.S who were subsequently excluded. Approximately 31.2% of our sample developed post-vaccination neurological complications, particularly with ChAdOx1nCov-19. A vulnerable clinical profile emerged, where over 40% of the symptomatic people showed comorbidities in their clinical histories. Defining the neurological risk profile, we found an increased risk for ChAdOx1nCov-19 of tremors (vs. BNT162b2, OR: 5.12, 95% CI: 3.51–7.48); insomnia (vs. mRNA-1273, OR: 1.87, 95% CI: 1.02–3.39); muscle spasms (vs. BNT162b2, OR: 1.62, 95% CI: 1.08–2.46); and headaches (vs. BNT162b2, OR: 1.49, 95% CI: 0.96–1.57). For mRNA-1273, there were increased risks of parethesia (vs. ChAdOx1nCov-19, OR: 2.37, 95% CI: 1.48–3.79); vertigo (vs. ChAdOx1nCov-19, OR: 1.68, 95% CI: 1.20–2.35); diplopia (vs. ChAdOx1nCov-19, OR: 1.55, 95% CI: 0.67–3.57); and sleepiness (vs. ChAdOx1nCov-19, OR: 1.28, 95% CI: 0.98–1.67). In the period that ranged from March to August 2021, no one was hospitalized and/or died of severe complications related to COVID-19 vaccinations. Discussion: This study estimates the prevalence and risk for neurological complications potentially associated with COVID-19 vaccines, thus improving the vaccination guidelines and loading in future personalized preventive medicine.

## 1. Introduction

The mass vaccination campaigns against COVID-19 posed emerging questions to clinicians on the risks, benefits and timing of vaccinations for a correct stimulation of the immune system. In the last year, four main vaccines against COVID-19 infection were developed and distributed worldwide. In detail, the ChAdOx1 nCov-19 (Oxford–AstraZeneca, Oxford, UK) and the Ad26.COV2.S (Janssen, Singapore) vaccine are recombinant adenoviral vectors and chimpanzee viral vector and human adenovirus type, respectively [1]. The BNT162b2 (Pfizer–BioNTech, New York, NY, USA) and mRNA-1273 (Moderna, Cambridge, MA, USA) are mRNA-based vaccines; the first works against the spike protein, while the second encodes a prefusion-stabilized full-length spike protein of the SARS-CoV2-virus [1]. Although the randomized clinical trials [2,3,4] have been confirmed by observational studies [5], and nationwide mass vaccination setting studies [6], the research has been questioned because of their lack of power to identify less common adverse events. This represents a limitation in the clinical trials, and this is why surveillance has continued after trials using observational data. An important scientific effort is now focused on identifying the specific safety profile for each vaccine. A recent nationwide mass vaccination setting study [7] demonstrated that BNT162b2 was associated with an excess risk of myocarditis.

When we examine under a magnifying glass the potential neurological complications post-vaccination, the first concern is dated from September 2020, when AstraZeneca/Oxford University reported severe inflammation of the spinal cord [8]. In May 2021, the American Neurology Academy provided the first report on the common neurological complications after COVID-19 vaccines [9]. In December 2021, a large population-based study identified rare neurological adverse events after the first dose of the ChAdOx1nCoV-19 and BNT162b2 vaccines [10]. Severe and unexpected post-vaccination complications have also recently been detected. They include cerebral venous sinus thromboses occurring especially in females after adenovector-based vaccination [11], as well as Guillain-Barré syndrome, facial palsy, other neuropathies, encephalitis, meningitis, myelitis and autoimmune disorders [12]. However, these events were found in isolated case reports or small series. On the other hand, it was demonstrated that the risk of severe neurological complications may be greater following a positive SARS-CoV-2 test [10]. Finally, a wide spectrum of mild, transient and self-limiting neurological complications such as fever and chills, headache, fatigue, myalgia and arthralgia, or local injection site effects has been also reported following COVID-19 vaccination [11]. However, little is yet known about the neurological complications after both doses of COVID-19 vaccines, as well as about their nature. Finally, the exact pathogenesis of vaccine-associated adverse events remains under debate. Several mechanisms based on the innate immune response and miming the reactivity to the viral infection have been recently proposed. This is the case for headaches that are secondary to SARS-CoV-2 infection [13,14]. Patients with long COVID headaches, especially in an acute phase of the disease, can manifest a persistent immune system activation with evidence of altered blood levels of cytokines and interleukins [13].

In this large population-based study, we aimed to evaluate the neurological complications after the first and second doses of COVID-19 vaccines approved in Italy (BNT162b2, mRNA-1273, ChAdOx1 nCov-19 and Ad26.COV2.S), and used in the massive vaccination hub of Novegro (Milan, Lombardy) between 7 and 16 July 2021. We tested the hypothesis that age, sex and vaccines can be useful to stratify the risk to develop adverse reactions in a large cohort of Italian people. Additionally, in a subpopulation of people developing neurological complications, we investigated the associations between vaccines and the development of specific adverse events. Finally, we evaluated whether the presence of comorbidities, as well as previous SARS-CoV-2 infection, can be useful to identify a clinical profile for those who are more vulnerable to develop neurological adverse effects.

## 2. Methods

### 2.1. Design, Setting and Participants

NEURO-COVAX is an observational study to identify acute and subacute neurological complications after (symptoms following vaccination) the administration of the first and second doses of COVID-19 vaccines used in the hub of Novegro (Milan, Lombardy) between 7 and 16 July 2021. We defined acute as neurological complications occurring within the first 15–30 min, and subacute as those occurring within the first two weeks after vaccination. Firstly, we collected the NEURO-COVAX questionnaires (details below) from the total group of participants during administration of the second dose (acute) of vaccines, with the possibility to retrospectively investigate the first dose-related effects (acute and subacute), and to prospectively evaluate the second dose (subacute-related effects). Secondly, we identified the symptomatic subgroup (subjects manifesting of at least one neurological symptom after at least one dose) to describe the neurological manifestations, onset and duration and the vaccine safety profile. To corroborate associations between vaccines and adverse events, we calculated the risk measures (odds ratios and relative risks). Thirdly, we characterized the symptomatic subgroup from a clinical point of view, identifying the following comorbidities or associated conditions: non-neurological disorders (cardiovascular diseases, lung diseases, kidney diseases, diabetes and blood diseases), neurological disorders (central and peripheral nervous system disorders), previous SARS-CoV2 infection, history of anticoagulants and tumoral drugs, pregnancy, breastfeeding and transfusions for hematological disorders. An accurate evaluation of immune system activity, hyperactivity (allergies—seasonal, food, materials, drugs; history of adverse reactions to previous vaccinations) and hyporeactivity (immunodeficiency conditions—leukemia, lymphoma, HIV and transplantations) was also performed. A total of 20.465 persons vaccinated in the Novegro hub during the period previously indicated were recruited for this population-based study. The eligibility criteria were the following: (i) an age of 18 years or older; (ii) COVID-19 vaccination performed at the hub of Novegro; and (iii) written informed consent for participation in the study. The hub of Novegro, managed by the Gruppo Ospedaliero San Donato, opened in April 2021 and closed in August 2021, was an excellent mass vaccination site in the Lombardy region (Italy). It represents an “immunization islands” model useful to improve the quality, efficiency and safety of COVID-19 mass vaccination sites [15]. In July 2021, four vaccine lines were available: BNT162b2 (80%), ChAdOx1nCov-19 (10%), mRNA-1273 (9%) and Ad26.COV2.S (1%). Data regarding the demographic data, anamnesis and COVID-19 infection were digitized centrally by the Lombardy region. The study was approved by the National Ethical Committee Spallanzani, National Institute for Infectious Diseases Lazzaro Spallanzani, Rome, Italy, under the project ID number 362 of the Trial Register 2020/2021, and by the local ethical committee of “Vita-Salute” San Raffaele University, Milan, Italy. Finally, all of our methods and experiments were performed in accordance with the relevant guidelines and regulations.

### 2.2. NEURO-COVAX Questionnaire

In our study, we were interested in developing a short measure to assess post-vaccination neurological complications. To this aim, we focused on the only literature available at the time the study was designed, in order to define the domain of observables. In this item generation stage, all of the possible post-vaccination complications listed in the Goss et al. [9] seminal study were considered and included in the first version of the questionnaire. Following feedback and revisions in item formulation received from 2 experienced clinicians (F.S.L and S.M.), minor item formulation revisions were considered and incorporated into the final version of the questionnaire.

Thus, the questionnaire was considered self-reported, consisting of 4 four simple sections: Section I—Vaccine Information: vaccine center, date of administration and vaccine type; Section II—Personal Data: surname and name; date of birth; telephone number/e-mail address and residence; Section III—Complications List: a grid listing the main adverse symptoms of our interest; Section IV—Complications Characterization: a box (symptom box) dedicated to each manifestation in which the participants could indicate the onset and duration of their symptoms. Combining the different timepoints of our evaluation and the number of doses planned for the vaccines, we designed 4 potential questionnaires for double dose vaccines and 2 for single dose-vaccines as follows: first dose acute; first dose subacute; second dose acute; second dose subacute; (single) dose acute; and (single) dose subacute. Thus, for each participant, we could potentially have more doses and more questionnaires (events), 1 person:2 doses:4 events, and we potentially expected to collect about 76.432 questionnaires (19.108 precipitants × 4 events). Of interest, under the section “other” of the questionnaire, the subject could note the following: (i) severe reactions not included in the questionnaire such as sudden loss of strength in the limbs, progressive difficulty walking and facial paralysis; (ii) access to emergency room; and (iii) hospitalizations in a neurological setting. A detailed description of the NEURO-COVAX questionnaire is reported in the Appendix A.

Regarding the questionnaire administration procedures, our protocol provided three steps. Firstly, the questionnaire was administered by the medical staff pre-vaccination during the anamnesis; secondly, during the waiting time post-vaccination, participants were asked to fill out the questionnaire on a voluntary basis. Highly trained and qualified medical staff (9–10 persons) explained how to fill in the questionnaire in the Italian language. Support was also provided in the English language for foreign people. Thirdly, after the questionnaires were filled in, the sections concerning the acute and subacute complications after the first dose and acute second dose were submitted to the staff, while the section that focused on the subacute complications of the second dose (or subacute single dose) required participants to send it to a dedicated hospital address via e-mail after the two vaccinations (neuro.covax@hsr.it). Participants could also be contacted via phone/e-mail by staff.

Finally, in order to identify the potential severe complications, such as Guillain-Barrè syndrome, Bell’s palsy, transverse myelitis, encephalitis and coagulation disorders, myocarditis, myocardial infarction, pulmonary embolism, pneumonia, that occurred after the observational period related to our questionnaire, we also performed accurate digital healthcare system matching to obtain data regarding incidences of subsequent hospital admission and/or mortality rates after the first and/or second dose of the vaccines. We evaluated these data for a period of up to 6 months (March–August 2021) to perform clinical observation that included both doses of the COVID-19 vaccines.

### 2.3. Statistical Analysis

A descriptive statistical analysis was conducted for each group, stratified according to the vaccine lines. Subjects receiving the Ad26.COV2.S vaccine were excluded from the statistical analysis because of a very limited sample number (n.12). We described the characteristics of each group in terms of sex and age, including the mean, standard deviation, standard error and median. Additionally, pre-established ranges were considered to investigate clinical variability across the age groups. Frequency analyses were implemented to assess the neurological complications and describe these in terms of onset and duration. We also calculated the event rates for the symptoms reported in our vaccinated cohort. In the total group, we constructed a multivariable logistic regression model to quantify associations between potential factors and adverse effects. In this multivariable model, the candidate factors were age, sex (female or all others) and vaccine lines. We used the presence of complications as the dependent variable, vaccine lines as factors and age as the covariate (continuous variable). A separate bivariate logistic regression model was constructed in the symptomatic group to identify adverse reactions associated to vaccine lines. The greater odds ratios (ORs) for each vaccine were presented, thus identifying the specific risk profile. Additionally, we used the chi-squared test for comparing the distribution of the neurological symptoms between the three groups of COVID-19-vaccinated people. The Bonferroni correction for multiple comparisons was applied to the *p* values. The significance level was set at *p* < 0.05. Finally, Fisher’s exact test was used to assess differences in positivity for the SARS-CoV-2 test among the three groups of vaccines. The significance level was set at *p* < 0.05. The relative risks (RRs) and confidence intervals (CIs) at 95% were calculated to evaluate the associations between dose/onset and the development of reactions. The first RR model was constructed considering the people exposed to the first dose compared to those exposed to the second dose. The RR was also calculated for each symptom and stratified according to the specific vaccine. The second RR model was constructed considering the people exposed that developed complications with acute onset compared to those with subacute onset. An RR greater than 1 indicated an increased risk to develop symptoms after the first dose and with an acute onset. A value of *p* < 0.05 was considered statistically significant, and all of the tests were 2-tailed. All of the analyses were performed with the use of JASP (version 0.15.0) and Jamovi (version 1.6.23), and re-tested with RStudio software (version 2021.09.02).

## 3. Results

### 3.1. Study Population

A total of 20,465 persons were considered for the current study. Of these, 19,108 participants (total group) were eligible for inclusion in the vaccination cohort. The remaining 1357 people were excluded because of incorrect completion of the questionnaire (656 incomplete, 397 lacked personal data, and 304 were not filled out) (Appendix A). The adherence to filling in the questionnaire was greater than 99% in the nine days of enrollment. The consent decline rate was less than 1%, considering that about half of the subjects who presented the uncompleted questionnaire did not also fill the written informed consent for participation in the study. For these reasons, these subjects were excluded from the study. We collected 67,340 (88.1%) questionnaires: 38,216 regarding the first dose of vaccines (19,108 acute and 19,108 subacute) and 29,124 regarding the second dose (19,108 acute and 10,016 subacute, equal to 52.4%). Table 1 summarizes the salient characteristics of all of the participants.

It is noteworthy that about a third of our sample developed manifestations after COVID-19 vaccinations (so-called symptomatic group). The demographic and clinical characteristics of this group are reported in Table 2. When we characterized the symptomatic group from a clinical point of view, some evidence emerged. Firstly, positivity to the SARS-CoV-2 test (4–6 months before vaccination) was significantly different for the comparison of the BNT162b2 vs. mRNA-1273 vaccine (*p* = 0.007) (Table 2). However, we did not exclude patients reporting no related SARS-CoV-2 symptoms who may have been positive for the PCR test. Indeed, only 5% of the participants performed SARS-CoV-2 testing, and had a negative result. In addition, these data could reflect the availability and utilization of COVID-19 testing at the time. Secondly, about 40% of the symptomatic group had specific comorbidities: allergies represented the largest share in all three of the vaccine brands. (Table 2). Finally, neurological diseases were less commonly found (3–6%) in symptomatic people who developed complications (Table 2). These findings corroborate our hypothesis, as the neurological adverse symptoms may be associated with COVID-19 vaccine rather than with previous neurological disorders.

### 3.2. Neurological Complications Post-Vaccination

The frequency distribution of the neurological complications after COVID-19 vaccines is detailed in Table 3 and schematically represented in Figure 1. Headaches had higher prevalence, especially for the mRNA-1273 vaccine. Sleep disorders including sleepiness and insomnia were frequent: sleepiness was the second most common neurological complication (37.8%), prevalently associated with the mRNA-1273 vaccine (39.7%), while insomnia was (5.1%) less common, and was more frequently associated with the ChAdOx1nCov-19 vaccine (5.8%). Cognitive fog (difficulty concentrating) was reported in less than 10% of our cohort, and mainly occurred after the mRNA vaccine, especially BNT162b2. Of note, we found 132 cases of diplopia (2.2%) and 115 cases of tinnitus (1.9%). While diplopia was slightly more frequent after the administration of an mRNA vaccine, especially mRNA-1273 (2.7% vs. 1.9%), tinnitus was more typically presented after an adenovirus vaccine. Smell and taste alterations were mainly reported in people reporting mRNA-1273. Table 4 shows the most frequent onset and duration of the neurological complications. Appendix A summarizes the characteristics of onset and duration for each neurological complication. The event rates of neurological symptoms were lower: headache (16%), sleepiness (11%), vertigo (4%), paresthesias (3%), cognitive fog (difficulty concentrating) (1.9%), insomnia (1.6%), muscle spasms (1.5%), tremor (1.5%), tinnitus (0.6%), taste alterations (0.3%), dysphonia (0.3%), smell alterations (0.1%). Overall, our results are in line with the last AIFA-Italian Medicines Agency report (N.13, 27 November 2020–26 September 2022), https://www.aifa.gov.it/documents/20142/1315190/Rapporto_sorveglianza_vaccini_COVID-19_13.pdf, accessed on 3 November 2022), as the most frequently reported neurological complications were the following: paresthesias, headaches and dizziness for the BNT162b2 and mRNA-1273 vaccines, and headache and sleepiness for the ChAdOx1 nCov-19 vaccine. Of note, we found novel neurological post-vaccination symptoms, which expanded the clinical spectrum related to COVID-19 vaccines. This could be mainly due to the different methods of investigation used. Indeed, in the AIFA report, it was required to spontaneously describe the case, whereas in our study it was necessary to define, in detail, the clinical features, onset and duration of each neurological manifestation indicated in the questionnaire.

Regarding the severe neurological complications, it is also necessary to emphasize that one participant (woman, BNT162b2) had access to the emergency room for sudden difficulty walking, one participant (man, BNT162b2) reported sudden memory loss (lasting 1 day) and one participant had facial paresis (young man, BNT162b2). Excluding these single cases, all of which occurred after the second dose of the BNT162b2 vaccine, we did not register from the self-reported questionnaires any severe adverse event in our vaccinated cohort. Additionally, the digital healthcare system matching also confirmed this, since none of the 19,108 subjects of our vaccinated cohort (symptomatic and not symptomatic) was hospitalized in neurological and/or non-neurological clinics and/or died in the observational period ranging from March to August 2021 for severe complications related to the COVID-19 vaccination.

### 3.3. Neurological Risk Profile

Regarding the risk profile, we observed an increased risk of neurological complications for the ChAdOx1nCov-19 (OR: 1.67, 95% CI: 1.34–2.06) vaccine, and a trend for the mRNA-1273 (OR: 1.12, 95% CI: 0.96–1.31) and BNT162b2 vaccines (OR: 0.91, 95% CI: 0.82–1.07). When comparing subjects vaccinated with BNT162b2, a significantly higher proportion of adverse neurological events was reported in people vaccinated with the ChAdOx1nCov-19 (*p* < 0.001) and mRNA-1273 vaccines (*p* < 0.001). We also found an increased risk of neurological complications in females (OR: 1.97, 95% CI: 1.80–2.09), whereas there was no risk association with the age (OR: 0.98, 95% CI: 0.98–0.99).

We defined a neurological risk profile for each vaccine line. We found increased risks for the ChAdOx1nCov-19 vaccine of tremors (vs. BNT162b2 OR: 5.12, 95% CI: 3.51–7.48); insomnia (vs. mRNA-1273, OR: 1.87, 95% CI: 1.02–3.39); tinnitus (vs. BNT162b2 OR: 1.75, 95% CI: 0.90–3.40); muscle spasms (vs. BNT162b2 OR: 1.62, 95% CI: 1.08–2.46); and headaches (vs. BNT162b2, OR: 1.49, 95% CI: 0.96–1.57). There was an increased risk for the mRNA-1273 vaccine of taste alterations (vs. ChAdOx1nCov-19: OR: 3.03, 95% CI: 0.83–10.7); parethesias (vs. ChAdOx1nCov-19: OR: 2.37, 95% CI: 1.48–3.79); smell alterations (vs. BNT162b2 OR: 1.96, 95% CI: 0.78–4.92); vertigo (vs. ChAdOx1nCov-19: OR: 1.68, 95% CI: 1–20-2.35); diplopia (vs. ChAdOx1nCov-19: OR: 1.55, 95% CI: 0.67–3.57); and sleepiness (vs. ChAdOx1nCov-19 OR: 1.28, 95% CI: 0.98–1.67). An increased risk of cognitive fog (difficulty concentrating) was reported for the BNT162b2 vaccine (vs. ChAdOx1nCov-19 OR: 1.54, 95% CI: 0.90–2.61).

We also observed an increased RR to develop almost all of the neurological complications after the first dose, and with an acute onset. There was an increased RR to develop almost all of the neurological complications after the first dose: sleepiness (RR = 2.20; 95% CI = 1.99–2.43); muscle spasms (RR = 1.67; 95% CI = 1.24–2.21); tremors (RR = 1.57; 95% CI = 1.19–2.08); taste alterations (RR = 1.51; 95% CI = 0.71–3.21); headaches (RR = 1.50; 95% CI = 1.41–1.59); smell alterations (RR = 1.45; 95% CI = 0.56–3.77); cognitive fog (difficulty concentrating) (RR = 1.42; 95% CI = 1.08–1.86); insomnia (RR = 1.25; 95% CI = 0.94–1.67); vertigo (RR = 1.22; 95% CI = 1.05–1.40); and tinnitus (RR = 1.14; 95% CI = 0.76–1.73). Meanwhile, a reverse trend was observed for diplopia (RR = 0.84; 95% CI = 0.60–1.17); dysphonia (RR = 0.84; 95% CI = 0.47–1.50); and paresthesia (RR = 0.80; 95% CI = 0.69–0.92). Figure 2 shows the risks of having specific adverse events after the first dose in comparison to the second, whereas Appendix A shows RR first vs. second dose, stratified for a single vaccine brand. Finally, we calculated the RR to develop neurological complications with acute onset according to the specific vaccine. Interestingly, we observed an increased risk to develop with acute onset almost all of the neurological complications: diplopia with mRNA-1273 (RR = 2.55; 95% CI = 1.27–5.09); paresthesia with mRNA-1273 (RR = 1.30; 95% CI = 0.57–2.08); tinnitus with mRNA-1273 (RR = 1.46; 95% CI = 0.58–3.68); sleepiness with ChAdOx1nCov-19 (RR = 1.21; 95% CI = 0.91–1.56); a trend for muscle spasms with ChAdOx1nCov-19 (RR = 1.10; 95% CI = 0.68–1.76); and tremors with ChAdOx1nCov-19 (RR = 1.10; 95% CI = 0.80–1.58). Of note, the clinical onset of cognitive fog, taste and smell alterations and insomnia was subacute for each vaccine and both doses. Figure 3 shows the RRs for single adverse neurological events stratified for each vaccine.

## 4. Discussion

This large population-based study in Italy investigated the neurological complications associated with first and second doses of three COVID-19 vaccines in use in the hub of Novegro (Milan, Lombardy). We identified several findings that were of clinical relevance to public health and scientific interest for clinicians and researchers. Firstly, we observed an increased risk of neurological adverse events in females, and for the adenovirus ChAdOx1nCov-19 vaccine, a trend for mRNA vaccines such as mRNA-1273 and BNT162b2. In line with this, a significant association between neurological symptoms following ChAdOx1nCov-19 and mRNA-1273 vaccination compared to BNT162b2 is also reported. Secondly, in the symptomatic vaccinated group, we identified a neurological risk profile that is specific for each vaccine. There is an increased risk for the ChAdOx1nCov-19 vaccine of tremors, insomnia, tinnitus, muscle spasms and headache; an increased risk for the mRNA-1273 vaccine of taste and smell alterations, vertigo, diplopia, sleepiness, parethesias and dysphonia; then, an increased risk for the BNT162b2 vaccine of cognitive fog. Finally, defining the symptomatic group, we found that over 40% of the subjects showed comorbidities in their clinical histories.

The neurological risk profile of the ChAdOx1nCov-19 vaccine included headaches, tremors, muscle spasms, insomnia and tinnitus. Of note, 53.2% (310/583) of people receiving the ChAdOx1nCov-19 developed headaches, with an increased risk after the first dose and subacute onset. Headaches had a short-term duration, being more frequently reversible within a day. Headaches have been reported as the third most common complication associated with vaccination against SARS-CoV-2. Results from a recent meta-analysis demonstrated that headaches occurred in 22% of the vaccinated people after the first and in 29% after the second vaccine dose over a 7-day period, with a higher percentage as compared to subjects receiving a placebo (10–12%) [16]. Headaches were also expected following the ChAdOx1nCov-19 vaccine. A clinical trial [17] found headaches to be the most common adverse event, probably related to the nature of this vaccine, which is a modified adenovirus vector vaccine; thus, it mimes not only the immunogenicity but also the side effects [18,19,20,21]. A fascinating hypothesis proposed that there is activation of the trigemino-vascular system mediated by the pathogen itself on trigeminal branches present at this level or through olfactory-trigeminal interactions, as the pathophysiological mechanism that underlies these headaches [22]. About 13.2% (77/583) of people receiving ChAdOx1nCov-19 developed tremors, with an increased risk after the first dose, and it commonly reverted within one day. Acute onset and short-term duration, however, seems to direct towards psychogenic rather than organic form. Psychogenic movement disorders especially tremor have been described in adolescents precipitating with H1N1 influenza vaccination [23]. New-onset movement disorders including psychogenic disorders have been found in COVID-19 cases [24]. Sleep disorders, in particular insomnia has been reported in 5.8% (34/583) persons receiving ChAdOx1nCov-19, with an increased risk after the first dose, and subacute onset and more frequently reversible risks within one week. Insomnia has been found in COVID-19 cases [25]; however, some questions remain unresolved. Firstly, we do not know whether our symptomatic people suffered from insomnia before vaccination; this may be of great interest, since insomnia is a risk factor that decreases the vaccine response [26,27]. Secondly, we do not know whether our symptomatic people actually developed an initial, intermediate or final insomnia rather than a misperception of their sleep quality due to vaccination stress. Future studies assessing sleep profiles before and after COVID-19 vaccination are needed. Tinnitus was reported in 2.7% (16/583) of people receiving ChAdOx1nCov-19, with increased risk after the first dose, and subacute onset and short-term duration; these features are suggestive of a transient phenomenon. Despite the incidence of tinnitus post-vaccination being infrequent, several cases have been described after COVID-19 vaccines [28,29]. A hypersensitivity reaction with an abnormal autoimmune response as cross-reactivity between anti-spike SARS-CoV-2 antibodies and otologic antigens or vasculitic event have been hypnotized to be pathogenetic mechanisms [28,29]. Overall, we can speculate that ChAdOx1nCov-19-related complications may be attributed to two main factors: firstly, the nature of the vaccine, which is a modified adenovirus vector that results in significant and persistent systemic immune activation; secondly, individual vulnerability related to a predisposing biology. Supporting this, higher serum levels of interleukin-6, angiotensin, ACE 2 as well as of lymphocytes have been detected in patients with long COVID infection with a predisposing headache biology [13].

The neurological risk profile of the mRNA-1273 vaccine included sleepiness, vertigo, diplopia, parethesias, taste and smell alterations, and dysphonia. Sleepiness was present in 39.7% (290/729) of people receiving mRNA-1273, with an increased risk after the first dose and acute onset. Its duration in most of the cases was within one week. Sleepiness in core adverse vaccination events is not unexpected. Firstly, an increased incidence of narcolepsy was observed in Europe following administration of Pandemrix, a vaccine against the H1N1 virus [30]. Secondly, a case of hypersomnia relapse after receiving the COVID-19 vaccine has been also reported: there was a case involving a female with a previous history of hypersomnia who presented excessive daytime sleepiness after a CoronaVac injection [31]. Finally, about 33.01% of the people affected by COVID-19 infections may experience excessive daytime sleepiness [32]. Taken together, this evidence suggests that there could be a strict relationship between the development of sleepiness and immune responses to vaccine/infection. A fascinating hypothesis suggests that influenza vaccines may lead to the selective immune-mediated destruction of orexin-producing neurons, which is T-cell-mediated neuronal damage, thus triggering narcolepsy [33,34]. Considering that the same can occur for COVID-19 vaccines, future investigations monitoring the new-onset hypersomnia findings in vulnerable individuals are urgently needed. About 15.9% (116/729) of people receiving mRNA-1273 showed vertigo, with an increased risk after the first dose and an acute onset. Its duration was more commonly within one day, suggesting a sudden but reversible complication. “Acute vertigo” post-COVID-19 vaccination has been reported [35]. An abnormal autoimmune response or a vasculitic event with subsequent localized damage to the cochlea have been proposed as potential mechanisms [36]. Parethesias were reported in 14.5% (106/729) of people receiving mRNA-1273, with an increased risk after the second dose and acute onset. These symptoms were more reversible after one day. A recent study [37] found non-specific sensory symptoms following BNT162b2 first-dose immunization, which is probably mediated by stress-related responses. On the other hand, cases of new-onset central nervous system demyelination, severe relapse in multiple sclerosis patients, as well as Guillain-Barrè syndrome have been reported [38,39,40]. These conditions, however, are not reversible in a short time period. Interestingly, diplopia has been reported in 2.7% (20/729) of people receiving mRNA-1273, with an increased risk after the second dose and acute onset. However, its short duration (within one day) was suggestive of a transient event. Binocular diplopia associated with a transient oculomotor nerve palsy occurred after the first dose of mRNA-1273 [41]. Isolated abducens nerve palsy may be present days after COVID-19 onset [42]. However, the immune response activated by a vaccine may have individual variability. A reactivation of the immune response and diabetes mellitus may be precipitant factors for oculomotor nerve palsy [41]. Our symptomatic people showed an increased risk to develop diplopia after the second dose, as if a reactivation of the immune response was necessary to trigger diplopia. Finally, taste/smell alterations may occur after mRNA-1273 with an increased risk after the first dose, subacute onset and a longer duration. Otolaryngology-specific symptoms have been described post-vaccination, especially in subjects with a previous COVID-19 infection [43].

Regarding the neurological risk profile of the BNT162b2 vaccine, we found an increased risk for cognitive fog (difficulty concentrating). Approximately 6.4% (300/4650) of people who received the BNT162b2 vaccine developed cognitive fog, with an increased risk of its presence after the first dose, subacute onset and reversibility within one week. Brain fog is a type of cognitive impairment that presents as a “foggy brain state”, including a lack of intellectual clarity, difficulty with concentration, mental fatigue and anxiety [44]. Hypotheses including systemic inflammation crossing the blood–brain barrier, neuroinflammation after viral infection leads and microglial activation are emerging as explanations of this phenomenon in COVID-19 patients [45]. An alternative speculation is that symptomatic people may have a subclinical cognitive dysfunction before vaccination, and that vaccination is a trigger. Prospective studies are required to investigate the relationship between the development of brain fog and vaccines. A previous nationwide observational prospective study performed in Mexico among 704,003 first-dose recipients of the BNT162b2 vaccine reported adverse events in 6536 vaccinated people (<1%); of these, 65.1% had at least one neurologic (non-serious 99.6%) event. Serious neurologic events were detected in 17 subjects, and in the presence of concomitant allergic, anaphylactic or infectious conditions [46]. Finally, the differences in terms of the frequency of clinical manifestations following the BNT162b2 vaccine reported in a nationwide mass vaccination setting [6] could be due to the sample size, methods of investigation and the characteristics of eligible populations.

It is also interesting to note that for any vaccine brand, excepting the cited single cases, we registered no severe neurological and/or non-neurological complications and/or death following COVID-19 vaccination. Moreover, no subject of our vaccinated cohort was hospitalized in a neurological and/or non-neurological setting during the six months of observation. This is not surprising, considering that the severe neurological complication post-vaccination are rare events, and an acute onset and an absolute risk for a causal effect attributable to the vaccination are low [47]. Similarly, recent evidence [48] found no association between mRNA-based vaccines and severe cardiovascular events including myocarditis. These findings are also in agreement with AIFA-Italian Medicines Agency report, where myocarditis was associated with the BNT162b2 and mRNA-1273 vaccines in 2 cases/1.000.000 and 4.5 cases/1.000.000 of administered doses, respectively (https://www.aifa.gov.it/documents/20142/1315190/Rapporto_sorveglianza_vaccini_COVID-19_13.pdf, accessed on 3 November 2022). Moreover, a low incidence of hospitalization with COVID-19 pneumonia or death following vaccination and booster with any of the BNT162b2, mRNA-1273 or Ad26.COV2.S vaccines was recently reported [49]. Taken together, our findings are highly reassuring, thus encouraging COVID-19 vaccination in the future.

Of note, we identified baseline factors that are potentially associated to adverse events. There is an increased risk of developing neurological complications in females. Our findings are in line with those of a recent study that revealed that several factors, including the female sex, were associated with greater odds of adverse effects [50]. Why this occurs is an intriguing question. Biologically, differences between females and males can affect COVID-19 infection, as well as contribute to sex-specific vaccine outcomes [51,52]. The mechanism is probably related to genetic and hormonal factors. A genetic profile characterizes the female sex, since the X chromosome contains the most prominent immune-related genes in the human genome [53], thus causing stronger inflammatory immune responses [54]. In addition, estradiol, a primary female sex steroid that binds to the cytoplasmic estrogenic receptors on T cells and B cells, triggers humoral immunity to produce antibodies against infections [55]. Furthermore, the evidence that immune system dysfunctions (allergies/immunodeficiency disorders) are frequently reported in our symptomatic group is more than a chance occurrence. Indeed, concerning the clinical profiles of the more vulnerable to develop post-vaccination complications, we reported the following: (i) 47.6% of the ChAdOx1nCov-19 vaccine symptomatic people showed comorbidities; allergies and non-neurological diseases have similarly been reported; a history of antitumoral and anticoagulant drugs was more frequent in this population; (ii) 38.8% of the mRNA-1273 vaccine symptomatic people showed comorbidities; allergies were more frequently represented (especially drugs); a history of neurological diseases and transfusions and previous SARS-CoV-2 was more frequently observed in this population; (iii) 41.5% of the BNT162b2 vaccine symptomatic people showed comorbidities, and allergies were more frequently reported (especially drugs). A history of immunodeficiency disorders was more commonly observed in this population.

This study has limitations. Firstly, our results should be interpreted with caution because of a possible overestimation of neurological events resulting from the self-reported symptoms. Secondly, we evaluated the risks associated with the first and second doses of the vaccine; however, the data concerning the second dose were limited, thus representing a potential bias in the study. Possible misunderstandings of the study and/or inabilities to send the subacute questionnaires may have occurred. However, considering the lack of further reports during the long period of time from the beginning of the study (July 2021), we can speculate that subjects who did not send questionnaires did not manifest neurological symptoms after the second dose. Thirdly, the variable composition of the eligible population regarding race, ethnicity, social status and education level may impact on the development of adverse events with different risk estimates. However, this study has several strengths. Firstly, this was a single-center population-based study that investigated neurological complications associated with the first and second doses of COVID-19 vaccines. Secondly, the study design allowed a serial neurological investigation at different time points, thus collection of prospectively recorded medical data up to an observational period of six months. Thirdly, the large sample size provided sufficient power to quantify the risks of neurological complications according to female sex, age, vaccine lines, doses and clinical onset, which were not assessed through clinical trials.

## 5. Conclusions

This study identified a specific neurological risk profile for each vaccine and a clinical profile for those more vulnerable to develop neurological complications after COVID-19 vaccines. Clinicians should be aware that several neurological complications may commonly occur after COVID-19 vaccines, but in most cases, these have a benign nature. On the other hand, caution should be used when administering COVID-19 vaccines to vulnerable people, such as to those who suffer from allergies. We strongly believe that our findings are relevant for public health regarding the safety of vaccines in a large cohort.

## Figures and Tables

**Figure 1 vaccines-11-01621-f001:**
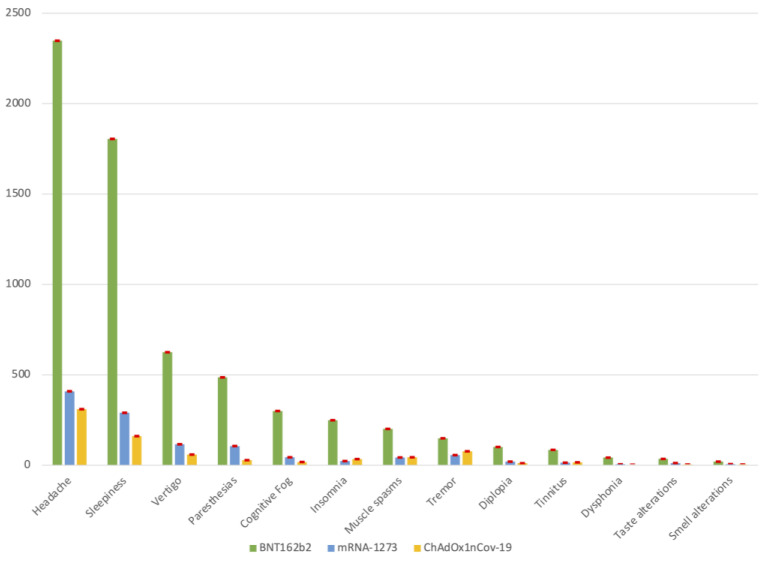
Neurological complications following COVID-19 vaccines. Distribution of adverse events, distinguished for specific symptom and stratified for each vaccine. Data are presented as total number of subjects presenting the specific neurological complication and standard error. The colors green, yellow and blue are representative of the BNT162b2, mRNA-1273 and ChAdOx1nCov-19 vaccines, respectively. The color red is representative of the standard error (SE). In detail, SEs for BNT162b2 vaccine are as follows: headache 0.50; sleepiness 0.49; vertigo 0.34; paresthesia 0.31; cognitive fog 0.25; insomnia 0.23; muscle spasms 0.20; tremor 0.18; diplopia 0.15; tinnitus 0.13; dysphonia 0.09; taste alterations 0.09; smell alterations 0.07. SEs for mRNA-1273 vaccine are as follows: headache 0.50; sleepiness 0.49; vertigo 0.37; paresthesia 0.35; cognitive fog 0.24; insomnia 0.17; muscle spasms 0.24; tremor 0.27; diplopia 0.16; tinnitus 0.14; dysphonia 0.08; taste alterations 0.13; smell alterations 0.09. SEs for ChAdOx1nCov-19vaccine were the following: headache 0.50; sleepiness 0.45; vertigo 0.30; paresthesia 0.21; cognitive fog 0.17; insomnia 0.23; muscle spasms 0.26; tremor 0.34; diplopia 0.14; tinnitus 0.16; dysphonia 0.07; taste alterations 0.08; smell alterations 0.08.

**Figure 2 vaccines-11-01621-f002:**
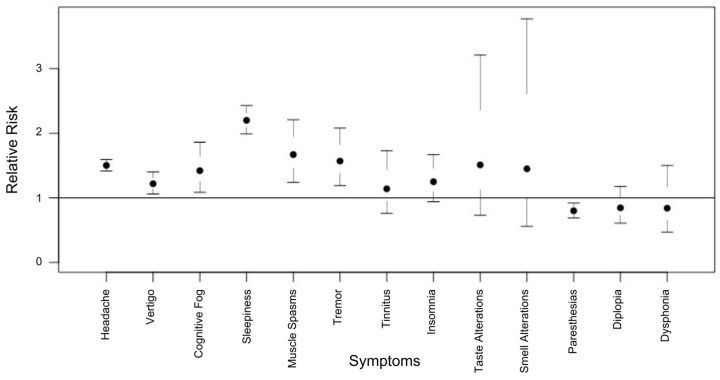
Relative risks (RRs) to develop neurological complications after first dose of COVID-19 vaccines. Increased RRs (>1) were observed for sleepiness (RR = 2.20; 95% CI = 1.99–2.43); muscle spasms (RR = 1.67; 95% CI = 1.24–2.21); tremors (RR = 1.57; 95% CI = 1.19–2.08); taste alterations (RR = 1.51; 95% CI = 0.71–3.21); headaches (RR = 1.50; 95% CI = 1.41–1.59); smell alterations (RR = 1.45; 95% CI = 0.56–3.77); cognitive fog (RR = 1.42; 95% CI = 1.08–1.86); insomnia (RR = 1.25; 95% CI = 0.94–1.67); vertigo (RR = 1.22; 95% CI = 1.05–1.40); and tinnitus (RR = 1.14; 95% CI = 0.76–1.73). Diplopia (RR = 0.84; 95% CI = 0.60–1.17); dysphonia (RR = 0.84; 95% CI = 0.47–1.50); and paresthesia (RR = 0.80; 95% CI = 0.69–0.92) showed a reverse trend (RR < 1).

**Figure 3 vaccines-11-01621-f003:**
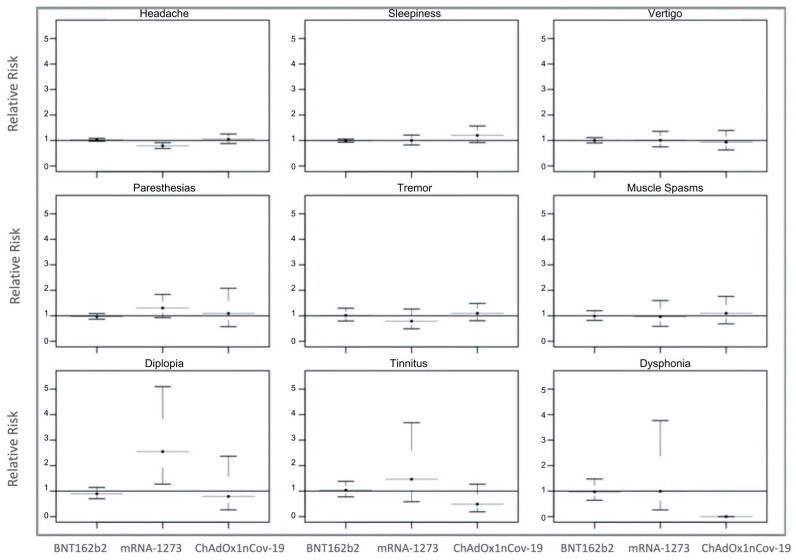
Relative risks (RRs) to develop neurological complications with an acute onset of COVID-19 vaccines. RRs were calculated for each symptom and stratified according to specific vaccine. Increased RR (>1) was observed for headaches with ChAdOx1nCov-19 (RR = 1.05; 95% CI = 0.88–1.25); diplopia with mRNA-1273 (RR = 2.55; 95% CI = 1.27–5.09); paresthesia with mRNA-1273; and ChAdOx1nCov-19 (RR = 1.30; 95% CI = 0.93–1.83; RR = 1.10; 95% CI = 0.57–2.08, respectively); tinnitus with mRNA-1273 (RR = 1.46; 95% CI = 0.58–3.68); sleepiness with ChAdOx1nCov-19 (RR = 1.21; 95% CI = 0.91–1.56); muscle spasms with ChAdOx1nCov-19 (RR = 1.10; 95% CI = 0.68–1.76); and tremors with ChAdOx1nCov-19 (RR = 1.10; 95% CI = 0.80–1.58). For all symptoms associated with a specific vaccine we observed an RR ≤ 1.

**Table 1 vaccines-11-01621-t001:** Demographic and clinical characteristics of the total group of people receiving COVID-19 vaccines in the massive Novegro hub (Milan, Lombardy) between 7 and 16 July 2021.

	BNT162b2 n(%)	Vaccine Lines	ChAdOx1nCov-19 n(%)
		mRNA-1273 n(%)	
Total number of people stratified for vaccines	15.368 (80.4)	2077 (10.8)	1651 (8.6)
**Sex**			
Female	7497 (48.7)	988 (47.5)	867 (52.5)
Male	7869 (51.2)	1089 (52.4)	784 (47.4)
Not Determined	2 (0.01)	-	-
**Age Groups (years)**			
Mean (s.d.)	45.9 ± 11.1	43.7 ± 10.8	66.1 ± 5.49
18–29	747 (4.86)	133 (6.40)	4 (0.24)
30–39	3834 (24.94)	764 (36.8)	12 (0.72)
40–49	4007 (26.07)	335 (16.1)	9 (0.55)
50–59	5736 (37.32)	799 (38.5)	23 (1.39)
60–69	684 (4.45)	31 (1.49)	1352 (81.89)
70–79	213 (1.39)	10 (0.48)	242 (14.66)
≥80	56 (0.37)	2 (0.09)	8 (0.49)
Not Determined	91 (0.6)	3 (0.14)	1 (0.06)
**Non-Neurological complications** **post-vaccination**			
Total number stratified for vaccines	553 (3.6)	131 (6.3)	171 (10.3)
**Neurological complications post-vaccination**			
Total number stratified for vaccines	4650 (30.2)	729 (35.1)	583 (35.3)

**Table 2 vaccines-11-01621-t002:** Demographic and clinical characteristics of the people receiving COVID-19 vaccines in the hub of Novegro (Milan, Lombardy) between 7 and 16 July 2021, developing neurological complications in at least one dose (symptomatic group).

	BNT162b2 (N = 4650)	Vaccine Lines	ChAdOx1nCov-19 (N = 583)
		mRNA-1273 (N = 729)	
**Sex n, (%)**			
Female	2788 (60)	439 (60)	380 (65)
Male	1862 (40)	290 (40)	203 (35)
**Age Groups (years)**			
Mean (s.d.)	46.0 ± 11.2	39.0 ± 11.2	66.0 ± 5.88
18–29	282 (6.06)	56 (7.68)	1 (0.17)
30–39	1340 (28.82)	315 (43.2)	4 (0.68)
40–49	1252 (26.92)	107 (14.6)	4 (0.68)
50–59	1495 (32.1)	232 (31.8)	9 (1.54)
60–69	192 (4.12)	16 (2.19)	487 (83.5)
70–79	44 (0.94)	2 (0.27)	78 (13.3)
≥80	15 (0.32)	1 (0.13)	-
Not Determined	30 (0.64)	-	-
**Positive SARS-CoV2 test (before vaccination) ^§^**	34 (0.73)	22 (3.0)	8 (1.37)
**Comorbidities or concomitant conditions**			
**Total Number n, (%)**	1933 (41.5)	283 (38.8)	278 (47.6)
Allergies ^x^	1310 (67.7)	184 (65.0)	140 (50.4)
Seasonal (pollen)	406 (31.0)	60 (32.6)	30 (21.4)
Food	389 (29.7)	67 (36.4)	38 (27.1)
Materials (latex)	55 (4.2)	6 (3.26)	4 (2.90)
Drugs	668 (51)	82 (44.5)	82 (58.6)
Cardiovascular diseases, lung diseases, kidney diseases, diabetes and blood diseases	600 (31.0)	98 (34.6)	138 (49.6)
Immunodeficiency disorders (leukemia, lymphoma, HIV and transplantation)	130 (6.72)	13 (4.59)	14 (5.03)
-Antitumoral Drugs	106 (5.48)	16 (5.65)	19 (6.83)
Neurological comorbidities (central and peripheral nervous system disorders)	114 (5.89)	17 (6.00)	9 (3.23)
Transfusions for hematological disorders	35 (1.81)	6 (2.12)	4 (1.43)
Pregnancy	36 (1.86)	7 (2.47)	-
Breastfeeding	33 (1.7)	5 (1.77)	
History of anticoagulant drugs	19 (0.98)	2 (0.70)	18 (6.47)
History of adverse reactions to previous vaccination	13 (0.67)	1 (0.35)	5 (1.79)

^x^ Data were calculated considering the total number of subjects with allergies, distinguished for vaccine brands. ^§^ BNT162b2 vs. mRNA-1273, *p* = 0.007; BNT162b2 vs. ChAdOx1nCov-19, *p* = 0.83; mRNA-1273 vs. ChAdOx1nCov-19, *p* = 0.14. Fisher’s exact test.

**Table 3 vaccines-11-01621-t003:** Frequency distribution of the neurological complications in the symptomatic group and distinguished according to vaccine lines.

Neurological Complications	Symptomatic Group ^x^ (N = 5962)	BNT162b2 (N = 4650)	Vaccine Lines ^x^	ChAdOx1nCov-19 (N = 583)
			mRNA-1273 (N = 729)	
Headache	3067 (51.4)	2348 (50.5)	409 (56.1)	310 (53.2)
Sleepiness	2256 (37.8)	1805 (38.8)	290 (39.7)	161 (27.6)
Vertigo	800 (13.4)	625 (13.4)	116 (15.9)	59 (10.1)
Paresthesia	620 (10.4)	486 (10.4)	106 (14.5)	28 (4.8)
Cognitive fog (difficulty concentrating)	362 (6.1)	300 (6.4)	44 (6.0)	18 (3.1)
Insomnia	306 (5.1)	249 (5.3)	23 (3.2)	34 (5.8)
Muscle spasms	288 (4.8)	201 (4.3)	43 (5.9)	44 (7.5)
Tremor	282 (4.7)	149 (3.2)	56 (7.7)	77 (13.2)
Diplopia	132 (2.2)	101 (2.2)	20 (2.7)	11 (1.9)
Tinnitus	115 (1.9)	85 (1.8)	14 (1.9)	16 (2.7)
Taste alterations	51 (0.9)	35 (0.7)	12 (1.6)	4 (0.7)
Dysphonia	50 (0.8)	42 (0.9)	5 (0.7)	3 (0.5)
Smell alterations	30 (0.5)	20 (0.4)	6 (0.8)	4 (0.7)

^x^ Data are expressed as numbers and percentages.

**Table 4 vaccines-11-01621-t004:** Clinical onset and durations of the neurological complications more frequently reported after COVID-19 vaccines.

Neurological Complication	First Dose		Second Dose	
	Clinical Onset (Minutes/Days)	Duration (Day/Week)	Clinical Onset (Minutes/Days)	Duration
Headache	Within first 15 min	1 day	Within first 3 days	1 day
Sleepiness	From 15 to 30 min	Up to a week	Within first 3 days	Up a week
Vertigo	From15 to 30 min	1 day	Within 30 min	1 day
Paresthesia	From 15 to 30 min	1 day	From 15 to 30 min	1 day
Cognitive fog	Within first 3 days	Up to a week	Within first 3 days	Up a week
Insomnia	Within first 3 days	Up to a week	Within first 3 days	Up a week
Tremor	Within first 15 min	1 day	Within first 3 days	1 day
Muscle spasms	Within first 15 min	<1 day	Within first 3 days	<1 day
Diplopia	From 15 to 30 min	1 day	Within first 15 min	1 day
Tinnitus	From 15 to 30 min	1 day	From 15 within 30 min	1 day
Taste alterations	Within 3 days	Up to a week	Within 3 days	Up a week
Dysphonia	Within first 3 days	<1 day	Within first 3 days	<1 day
Smell alterations	Within 3 days	>a week	Within 3 days	>a week

## Data Availability

The data that support the findings of this study are available on reasonable request from the corresponding author. The data are not publicly available due to privacy or ethical restrictions.

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
