# Peer review of "NEURO-COVAX: An Italian Population-Based Study of Neurological Complications after COVID-19 Vaccinations"

_vaccines, 2023, doi:10.3390/vaccines11101621_

Round 1

Reviewer 1 Report

The paper presented to me for review deals with a very important issue, which is the risk of neurological complications in patients vaccinated against COVID-19. The paper is based on NEURO-COVAX-cohort included 19,108 vaccinated people. The authors showed that about 31.3% of sample developed post-vaccination neurological complications, particularly ChAdOx1nCov. Most were not serious complications. None was hospitalized and/or died for severe complications related to COVID-19 vaccination. 

The paper is based on a very large body of material, written in clear and understandable language, and the extensively discussed methodology is not objectionable. However, before accepting the paper for publication, I would suggest several additions:

1. since the mechanism of neurological symptoms after vaccination is unclear, it is believed that it may be similar to the immune response as in COVID-19 infection itself, it is worthwhile to complete this in the introduction based on: PMID: 35758225 and PMID: 35915417

2. in the discussion when the incidence of headaches is analyzed, it would be appropriate to refer to the largest meta-analysis on the incidence of headaches after vaccination conducted on 1.57million patients: PMID: 35361131

Author Response

Reviewer’s comment: The paper presented to me for review deals with a very important issue, which is the risk of neurological complications in patients vaccinated against COVID-19. The paper is based on NEURO-COVAX-cohort included 19,108 vaccinated people. The authors showed that about 31.3% of sample developed post-vaccination neurological complications, particularly ChAdOx1nCov. Most were not serious complications. None was hospitalized and/or died for severe complications related to COVID-19 vaccination.

The paper is based on a very large body of material, written in clear and understandable language, and the extensively discussed methodology is not objectionable. However, before accepting the paper for publication, I would suggest several additions.

Authors’ response: we thank the Reviewer for appreciating our paper and for her/his comment. Thus, we have the opportunity to improve it further.

Reviewer’s comment: Since the mechanism of neurological symptoms after vaccination is unclear, it is believed that it may be similar to the immune response as in COVID-19 infection itself, it is worthwhile to complete this in the introduction based on: PMID: 35758225 and PMID: 35915417.

Authors’ response: According to the Reviewer’s comment, we added some sentences on this interesting topic in the introduction (page 4, lines 86-91) and references in the reference’s list (page 19, lines 516-519).

Reviewer’s comment: In the discussion when the incidence of headaches is analyzed, it would be appropriate to refer to the largest meta-analysis on the incidence of headaches after vaccination conducted on 1.57million patients: PMID: 35361131.

Authors’ response: We thank the Reviewer for her/his correct comment. In the new version of the manuscript, we have better specified the post-vaccination incidence of headache (page 12, lines 323-326) and added the reference in the reference’s lists (page 19, lines 522-524).

Reviewer 2 Report

In the current paper, the authors presented the results of the NEURO-COVAX-cohort study, a single-center Italian study exploring neurological complications of COVID-19 vaccination. They reported post-vaccination neurological complications in about one-third of cases, especially following the AstraZeneca vaccination. Overall, the findings are intriguing. The novelty of the presented results is a crucial strength point. However, I have some important concerns that the authors should consider before publication.  

In the abstract, the authors explain that 12 subjects had the Ad26.COV2 and were "subsequently excluded," but there is no explanation in the text about the exclusion.

The introduction can be expanded adding clarity to the manuscript. I suggest the following review on this topic: doi:10.1007/s10072-021-05662-9 and doi: 10.3390/life12091338. In addition, I suggest an overview of the neurological complications of the COVID-19 vaccine and also regarding COVID-19 infection (see doi: 10.1038/s41591-021-01556-7). Lastly, the introduction would benefit from a more detailed description of vaccine types, isoforms, doses, and administration time. Consider, for clarity, using consistent names for the vaccines (the authors use indifferently molecular and brand names of vaccines - for instance, Pfizer is named for the first time in the results section. Please show consistency.

The figures in the supplementary need revision since there are some typos, especially in the flowchart. In the methods, please specify why the authors defined complications within 15-30 minutes as acute and two weeks after vaccination as subacute, possibly including a reference. In addition, the authors should specify why they chose those symptoms reported in the questionnaire and how comorbidities were selected - please also correct the sentence "heart, lung, kidney asthma, diabetes"; the reported comorbidities in the methods are fewer than those reported in table 2; some information reported in the results are missed in the methods, including the presence of a positive SARS-COV2 test.

In the results section, the sentence "Of note, we collected 67.324 (88.1%) questionnaires, as follows: 19.108 first dose acute 19.108 second dose subacute, 19.108 second dose acute and 10.016 (52.4%) second dose subacute." is unclear, please correct; in the discussion section, the authors should provide a brief comment on the reduction of about 50% of responses about the second dose subacute effect.
The section regarding the presence of a positive SARS-COV2 test before the vaccination needs to be clarified: how many people performed the test? Is the positivity to the test significantly different between groups of vaccines? If so, please report the p-value, also in table 2.

As a confirmation of the confusion regarding vaccine naming, the name "INN-covid 19 mRNA" appears in page 7 at first, without any explanation. The percentages about sleepness are inconsistent between the text and the table. Please correct 1.88 by 1.9. A more detailed explanation for data reported in table 4 is needed since it is unclear the meaning of timing and ranges (do the columns indicate means? range?). In the results, the classification of acute/subacute symptoms has been completely neglected: if there is a reason for this choice, the authors should explain it.

The discussion makes it unclear how the authors explain differences in neurological manifestations between vaccine groups. For instance, Headache is discussed only for AstraZeneca vaccine, despite being present in the other groups, and the authors do not provide a hypothesis about the different profiles.  
The sentence "Despite the neurological profile of Pfizer vaccine included several neurological symptoms, we found an increased risk for cognitive fog (difficulty concentrating)." is confusing. Please rephrase.
The discussion should include an essential report about neurological complications of covid19 vaccination (doi:  10.1016/j.clim.2021.108786)

Regarding the absence of severe neurological manifestations in the enrolled sample, the authors should consider (and discuss) a potential bias. The authors enrolled subjects while administering the second dose of the vaccine. However, some subjects might have been accidentally excluded. In fact, subjects suffering from severe adverse effects after the first dose could not undergo the second dose of the vaccine. Please discuss this point.            

Minor editing

Author Response

Reviewer’s comment: In the current paper, the authors presented the results of the NEURO-COVAX-cohort study, a single-center Italian study exploring neurological complications of COVID-19 vaccination. They reported post-vaccination neurological complications in about one-third of cases, especially following the AstraZeneca vaccination. Overall, the findings are intriguing. The novelty of the presented results is a crucial strength point. However, I have some important concerns that the authors should consider before publication. 

Authors’ response: we thank the Reviewer for the opportunity to improve our manuscript.

Reviewer’s comment: In the abstract, the authors explain that 12 subjects had the Ad26.COV2.S and were "subsequently excluded," but there is no explanation in the text about the exclusion.

Authors’ response: It is correct. In the new version of the manuscript, we provided an explanation for this exclusion (page 7, lines 178-180).

Reviewer’s comment: The introduction can be expanded adding clarity to the manuscript. I suggest the following review on this topic: doi:10.1007/s10072-021-05662-9 and doi: 10.3390/life12091338. In addition, I suggest an overview of the neurological complications of the COVID-19 vaccine and also regarding COVID-19 infection (see doi: 10.1038/s41591-021-01556-7). Lastly, the introduction would benefit from a more detailed description of vaccine types, isoforms, doses, and administration time. Consider, for clarity, using consistent names for the vaccines (the authors use indifferently molecular and brand names of vaccines - for instance, Pfizer is named for the first time in the results section. Please show consistency.

Authors’ response: We thank the Reviewer for her/his stimulating comment. Now, we have added some sentences in the Introduction (page 3, lines 60-65; pages 3/4, lines 78-86) and references in the references list’s (page 18, line 488-489; page 19, lines 512-515). We also modified the denomination of vaccines in the manuscript.

Reviewer’s comment: The figures in the supplementary need revision since there are some typos, especially in the flowchart. In the methods, please specify why the authors defined complications within 15-30 minutes as acute and two weeks after vaccination as subacute, possibly including a reference. In addition, the authors should specify why they chose those symptoms reported in the questionnaire and how comorbidities were selected - please also correct the sentence "heart, lung, kidney asthma, diabetes"; the reported comorbidities in the methods are fewer than those reported in table 2; some information reported in the results are missed in the methods, including the presence of a positive SARS-COV2 test.

Authors’ response: We thank the Reviewer for her/his punctual comment. Here, we have modified the Supplementary Figures. The choice to distinguish the adverse events into acute (15-30 minutes) and subacute (the first hours, first 3 days, from the 4th to the 7th day and, from the 8th to the 14th day), was based on the guidelines of National Health System (acute:20 minutes) and our clinical experience performed in the massive vaccination Hub Novegro. The comorbidities reported in Methods and Table 2, have been selected on the basis of those collected in the vaccination schedule approved by our National Health System, as well as, the information about the presence of a positive SARS-CoV-2 test. Regarding all other issues, please see page 5, lines 116-122; pages 5/6 137-142.

Reviewer’s comment: In the results section, the sentence "Of note, we collected 67.324 (88.1%) questionnaires, as follows: 19.108 first dose acute 19.108 second dose subacute, 19.108 second dose acute and 10.016 (52.4%) second dose subacute." is unclear, please correct; in the discussion section, the authors should provide a brief comment on the reduction of about 50% of responses about the second dose subacute effect.

Authors’ response: We have modified the sentence (page 8, lines 215-217) and added a comment in the limitation section (page 16, lines 444-447).

Reviewer’s comment: The section regarding the presence of a positive SARS-COV2 test before the vaccination needs to be clarified: how many people performed the test? Is the positivity to the test significantly different between groups of vaccines? If so, please report the p-value, also in table 2.

Authors’ response: We thank the Reviewer for her/his correct comment. Comparing the positivity to the test among the three vaccine brands, p-value was significantly different for BNT162b2 vs mRNA-1273 (p=0.007, Fisher’s exact test). We also added some sentences in the manuscript (page 7, line 192-194; page 8, lines 222-224) and p-values for all comparisons in Table 2.

Reviewer’s comment: As a confirmation of the confusion regarding vaccine naming, the name "INN-covid 19 mRNA" appears in page 7 at first, without any explanation. The percentages about sleepiness are inconsistent between the text and the table. Please correct 1.88 by 1.9. A more detailed explanation for data reported in table 4 is needed since it is unclear the meaning of timing and ranges (do the columns indicate means? range?). In the results, the classification of acute/subacute symptoms has been completely neglected: if there is a reason for this choice, the authors should explain it.

Authors’ response: We have modified the manuscript, according to the Reviewer’s suggestion (page 9, line 236-237). We added Supplementary Table 1 to better define onset and duration of neurological complications (page 9, lines 244-245),

Reviewer’s comment: The discussion makes it unclear how the authors explain differences in neurological manifestations between vaccine groups. For instance, Headache is discussed only for AstraZeneca vaccine, despite being present in the other groups, and the authors do not provide a hypothesis about the different profiles. 

Authors’ response: We thank the Reviewer for the possibility to better explain this point. It is correct. We identified the neurological risk profile for each vaccine on the basis of greater OR values associated to the specific symptom. For example, although headache was present both in BNT162b2 vs mRNA-1273 vaccine,

ChAdOx1nCov-19 showed a greater OR: 1.49. We added this point in the methods (page 7, lines 189-190).

Reviewer’s comment: The sentence "Despite the neurological profile of Pfizer vaccine included several neurological symptoms, we found an increased risk for cognitive fog (difficulty concentrating)." is confusing. Please rephrase. The discussion should include an essential report about neurological complications of covid19 vaccination (doi:  10.1016/j.clim.2021.108786).

Authors’ response: Now, we have rephrased the sentence (page 14, lines 390-391). We also discuss the cited report (page 15, lines 399-403) and add the reference in the reference’s list (page 22, lines 588-590).

Reviewer’s comment: Regarding the absence of severe neurological manifestations in the enrolled sample, the authors should consider (and discuss) a potential bias. The authors enrolled subjects while administering the second dose of the vaccine. However, some subjects might have been accidentally excluded. In fact, subjects suffering from severe adverse effects after the first dose could not undergo the second dose of the vaccine. Please discuss this point.           

Authors’ response: We share the concern of the Reviewer. However, we feel to be reassuring on this point. In our vaccinated cohort, only 3 subjects showed severe neurological complications, all after the second dose of BNT162b2 vaccine (difficulty walking, sudden memory loss, facial paresis) (page 10, line 260). Excluding these single cases, we did not register from any severe adverse events following the first dose of vaccines, both from self-reported questionnaires, as well as, from the digital healthcare system matching.

Reviewer 3 Report

The manuscript titled NEURO-COVAX: An Italian Population-Based Study of Neurological Complications after COVID-19 Vaccines presented by Maria Salsone et al is an excellent work, the result of a great effort

It is pleasant to read, very clear and explained, just some style comments:

• There are some typographical errors in the manuscript, please review it carefully (for example: Page 4 line 178: groups of COID-19 vaccinated people)

• Tables: organize the format of the tables better so that they are clearer, for example the legend of vaccines in table 2 is wrong, also in this table in the allergens use the tabulator to indicate that after indicating the total, the The following lines correspond to the most common

• In figure 1 you could put total numbers and their standard error instead of percentages

• Page 10 from line 293 to 300 the text formatting is not justified

• The discussion is broad and specific, but I need to address something else, for example if the AstraZeneca vaccine, its design may have a relationship with the side effects observed (chimpanzee adenovirus) or that the other RNA vaccines, due to their structure, may interact with target sites in the CNS leading to adverse effects

Congratulations for the excellent work

Author Response

Reviewer’s comment: The manuscript titled NEURO-COVAX: An Italian Population-Based Study of Neurological Complications after COVID-19 Vaccines presented by Maria Salsone et al is an excellent work, the result of a great effort

It is pleasant to read, very clear and explained, just some style comments.

Authors’ response: we thank the Reliever for appreciating our paper and for her/his comment thus providing to improve it further.

Reviewer’s comment There are some typographical errors in the manuscript, please review it carefully (for example: Page 4 line 178: groups of COID-19 vaccinated people).

Authors’ response: It is correct. We have revised our manuscript according to the Reviewer’s suggestion.

Reviewer’s comment: Tables: organize the format of the tables better so that they are clearer, for example the legend of vaccines in table 2 is wrong, also in this table in the allergens use the tabulator to indicate that after indicating the total, the following lines correspond to the most common.

Authors’ response: Now, we have modified Table 2, as suggested by the Reviewer.

Reviewer’s comment: In figure 1 you could put total numbers and their standard error instead of percentages

Authors’ response: Now, we have modified Figure 1, as suggested by the Reviewer.

Reviewer’s comment: Page 10 from line 293 to 300 the text formatting is not justified

Authors’ response: in the new version of the manuscript, we have modified the formatting of the text.

Reviewer’s comment: The discussion is broad and specific, but I need to address something else, for example if the AstraZeneca vaccine, its design may have a relationship with the side effects observed (chimpanzee adenovirus) or that the other RNA vaccines, due to their structure, may interact with target sites in the CNS leading to adverse effects.

 Authors’ response: we thank the Reviewer for her/his interesting comment. Now we added some sentences in the discussion (page 13, lines 349-354).

Round 2

Reviewer 2 Report

The authors replied to all my concerns. The clarity of the manuscript has been improved. I suggest the authors to correct few mistakes

Please correct inconsistency with the use of vaccines name (for example, avoid the use of “ChAdOx1 nCov-19”, which is used a few times rather than “ChAdOx1nCov-19”)

Please correct COVID19 with COVID-19

Minor check

Author Response

Reviewer #2:

Reviewer’s comment: The authors replied to all my concerns. The clarity of the manuscript has been improved. I suggest the authors to correct few mistakes

Authors’ response: we thank the Reviewer for appreciating our revision work, and for the opportunity to improve further our manuscript.

Reviewer’s comment: Please correct inconsistency with the use of vaccines name (for example, avoid the use of “ChAdOx1 nCov-19”, which is used a few times rather than “ChAdOx1nCov-19”)

Authors’ response: We have corrected the vaccine name in the Supplementary Table 1.

Reviewer’s comment: Please correct COVID19 with COVID-19

Authors’ response:  According to the Reviewer’s comment, here we have
